# Creativity in Recreational Figure Roller-Skating: A Pilot Study on the Psychological Benefits in School-Age Girls

**DOI:** 10.3390/ijerph191811407

**Published:** 2022-09-10

**Authors:** Juan Manuel García-Ceberino, Sebastián Feu, María Gracia Gamero, Santos Villafaina

**Affiliations:** 1Facultad de Educación y Psicología, Universidad de Extremadura, 06006 Badajoz, Spain; 2EMOTION Research Group, University of Huelva, 21071 Huelva, Spain; 3Optimization of Training and Sports Performance Research Group (GOERD), University of Extremadura, 10003 Cáceres, Spain; 4Facultad de Ciencias del Deporte, Universidad de Extremadura, 10003 Cáceres, Spain; 5Departamento de Desporto e Saúde, Escola de Saúde e Desenvolvimento Humano, Universidade de Évora, 7004-516 Évora, Portugal

**Keywords:** basic psychological need, creative intervention, emotion, skater, sports adherence

## Abstract

Creative strategies allow students to feel ownership of their learning, fostering interest and motivation towards sports and educational contexts. This study aimed to compare different psychological variables after applying creative and traditional sessions of recreational figure roller-skating. Twelve school-age female skaters (9.00 ± 1.09 years old) participated in this pilot study. They performed two sessions: (1) a creative session (where participants created their own choreography) and (2) a traditional session (where participants followed the choreography created by the sports professional). In the creative session, participants created their choreographies without instructions. The basic psychological needs scale, the measure of intentionality to be physically active (sports adherence) and the games and emotions scale were administered after each session. The creative intervention led to a higher satisfaction of the needs of perceived competence (*p*-value = 0.04; effect size = 0.59), social relationships (*p*-value = 0.03; effect size = 0.62) and adherence to figure roller-skating (*p*-value = 0.02; effect size = 0.69), compared to the traditional intervention in female skaters. Participants showed significantly more humor and less surprise in the creative session than in the traditional session. This greater satisfaction with perceived competence and social relationships could translate into greater adherence to sports.

## 1. Introduction

Physical activity behavior is an essential lifestyle determined by personal, social and environmental factors [1]. In this regard, physical activity performed during leisure time is related to the better social health status of children [2], reducing the risk of developing chronic diseases, such as cardiovascular diseases [3], diabetes [4], cancer [5], or depression [6]. For children and youth (ages 5 to 17 years), the World Health Organization [7] recommends at least 60 minutes of daily moderate to vigorous physical activity, mainly aerobic exercises. Thus, higher levels of physical inactivity have been highlighted as one of the major problems in current society [8,9], leading to a significant increase in obesity worldwide [10].

Strategies, such as active play, which increases physical activity and prevents childhood obesity [11], or out-of-school sports activities in public institutions and private clubs, have been proposed for children and youth [12]. These activities have positively affected health status or education [13]. In Europe, clubs and public institutions offer different sports modalities, including figure skating. This sport modality is complex because it contains specific characteristics for school-age athletes: high physical ability levels, artistic and aesthetic sense of movements and good spatial orientation [14]. These characteristics could make figure skating, although attractive, not among the recreational preferences of children and youth [15]. However, figure skating practice contributes to forming proactive behaviors, and improving mental and social skills [15].

The type of methodological approach used during sports intervention will determine the degree of enjoyment, satisfaction and sports adherence [16]. In this line, previous studies [17,18,19,20] have demonstrated the benefits of using innovative and creative interventions with school-aged children. Therefore, it is necessary to develop creative thinking in educational environments (school and out-of-school) through creative interventions [18]. This type of intervention could significantly impact the satisfaction of basic psychological needs (BPNs), which are associated with sports adherence [16,21]. The BPNs theory proposes that autonomy, competence and social relationships should be satisfied to increase self-determined motivation and promote sports adherence, well-being and health. When BNPs are not satisfied, demotivation and subsequent abandonment of sports practice [22,23] could occur. In addition, emotions induced by sports are also associated with sports adherence [24]. Thus, the study of emotions induced by sports sessions is a topic of great importance for designing, implementing and evaluating programs in the field of physical education and sports [25].

Sierra-Díaz et al. [19] encouraged further studies investigating and summarizing the results of innovative and creative interventions on psychosocial variables and sports adherence compared to traditional strategies. Therefore, this study aimed to investigate the impact of creative and traditional figure roller-skating sessions on the satisfaction of BPNs, their intentionality to be physically active and their emotions. We hypothesized that: (1) BPNs would be more satisfied through the creative session; (2) a creative session will result in greater adherence to skating than traditional intervention; and (3) creative interventions will result in greater positive emotions. In contrast, a traditional session will result in greater negative emotions.

## 2. Materials and Methods

### 2.1. Participants

A total of 12 girls aged 7–11 years old (9.00 ± 1.09 years old) with previous figure roller-skating experience of 4.33 ± 1.50 years participated in this pilot study. Participants who were measured in this pilot study were enrolled in recreational figure roller-skating for 8 months, from October to May 2022, in a public institution in the southwestern part of Spain. During the season, participants learned the basic elements of figure roller-skating, such as spins, basic jumps, pirouettes and artistic figures. The inclusion criteria were: (1) to practice recreational figure roller-skating during the entire sport season (October to May) and (2) to be able to read, understand and answer the questionnaires presented in this study.

The present study respected the ethical guidelines of the Helsinki Declaration of 1975 (with modifications in subsequent years) and the Organic Law 3/2018, of 5 December, on the protection of personal research data and the guarantee of digital rights (BOE, 294, 6 December 2018) to guarantee the ethical considerations of scientific research with human beings. The university’s Bioethics Committee approved all the procedures conducted in this study [protocol code: 105/2022]. Before data acquisition, the public institution and the sports professional gave their consent to develop the study. Then, parents, legal guardians and roller-skaters were informed. Lastly, legal guardians or parents gave written informed consent.

### 2.2. Variables and Instruments

All the questionnaires employed in this study were adapted for the school-age population. The dependent variables and questionnaires used in this study were:Satisfaction of BPNs. The Spanish version of the BPNs scale [26] was used. Each dimension/need (i.e., autonomy, perceived competence and social relationships) is composed of four items that are answered on a Likert-type scale of 1–5 points, where 1 = strongly disagree and 5 = strongly agree. Cronbach’s alpha of the instrument was excellent for the creative (*α* = 0.90) and traditional (*α* = 0.95) interventions.Sports adherence. The Spanish version of the measure of intentionality to be physically active (MIFA) instrument [27] was used. It is composed of five items. The type of scale and response range is similar to the one used on the BPNs scale. Cronbach’s alpha of the instrument was good for the creative intervention (*α* = 0.87) and acceptable for the traditional intervention (*α* = 0.79).Emotions. The games and emotions scale (GES) [25] was used. It assesses the participants’ scores from 0 to 10, corresponding to 13 emotions, after practicing sports activities. Emotions are classified as: four positive emotions (joy, humor, love and happiness), six negative emotions (fear, anger, rejection, sadness, shame and anxiety) and three ambiguous/neutral emotions (hope, compassion and surprise).

Moreover, qualitative information regarding participants‘ opinions was recorded during a final reflection after both sessions using a voice recorder. Roller-skaters were asked: What did they like most about the session conducted? Why? Additionally, a semi-structured interview was conducted with the sports professional. He answered the following questions: What were the skaters’ technical levels? What aspects would you highlight from the two sessions?

The independent variables were the creative and traditional interventions. The characteristics of each of the interventions are shown in Table 1.

### 2.3. Procedure

Both sessions were performed two weeks apart during the season’s last month. The decision of whether to do the creative or traditional session first was made via lottery. In this regard, the first intervention was the creative session, in which the participants were grouped into sub-groups and created their own choreographies. The second session was the traditional session, in which participants learned and performed a choreography following the sports professional’s guidelines. Both sessions were conducted by the same sports professional.

#### 2.3.1. Creative Session

This session proceeded as follows: (1) Warm-up using a non-specific game; (2) formation of sub-groups and creation of the choreographies; (3) practice the choreographies before presenting them to the other peer groups; (4) final exhibition of the choreographies; (5) answer the instruments; and (6) final reflection and cool-down using stretches. The duration of the session was one hour. Following Lipman’s [29] guidelines, the creative intervention contained nine characteristics:Productive: ability to create choreographies.Imaginative: use of imagination in their creation.Independent: cooperative work with the guidance of the sports professional.Experimentation: practice the choreographies created before the final exhibition in front of peer groups.Holism: incorporating all the learning achieved during the training season and new ideas into the choreographies.Expression: creativity is manifested.Transcendence: participants surpass themselves when executing spins, basic jumps, pirouettes and artistic figures.Surprise and amazement: incorporate novel and imaginative variations/elements that generate amazement.Generativity: variations / elements are adapted to the participant’s capabilities.

#### 2.3.2. Traditional Session

Two weeks later, the participants learned and performed a choreography following the sports professional’s guidelines. The traditional session proceeded as follows: (1) Warm-up using a non-specific game; (2) practice the choreography according to the established guidelines; (3) final exhibition of the choreography to be observed and evaluated by the sports professional; (4) answer the instruments; and (5) final reflection and cool-down using stretches. The duration of the session was one hour.

### 2.4. Statistical Analysis

The Statistical Package for Social Sciences, version 25 (IBM Corp. Released 2017. IBM SPSS Statistics for Windows, Version 25, IBM Corp, Armonk, NY, USA), was used to conduct the statistical analyses. The significance level was set at a *p*-value < 0.05. A Shapiro–Wilk test was performed, and according to the obtained results and the relatively low sample size, non-parametric analyses were performed.

One participant did not attend the creative session, and another did not attend the traditional session. However, data from the 12 initial participants were included in the analysis after conducting an intention-to-treat analysis by multiple imputations (MI) of missing values. Sterne et al. [30] guidelines were followed, and data was classified as missing at random. The SPSS software was used for MI of data.

Differences between creative and traditional sessions were explored using the Wilcoxon signed-rank test for satisfaction of BPNs, sports adherence and emotions. The effect sizes [*r*] were calculated for each comparison as well as classified using the following criteria: <0.1 as a small effect, between 0.1 and 0.5 as a medium effect and >0.5 as a large effect [31,32].

## 3. Results

### 3.1. Quantitative Results

Table 1 shows the descriptive data of the BPNs (autonomy, competence and social relationships), MIFA (adherence to sports) and GES (emotions) according to the type of intervention. The creative intervention in figure roller-skating showed higher satisfaction in all these psychological variables compared to the traditional intervention. In addition, the creative intervention promoted more positive emotions in participants than the traditional intervention, while the traditional intervention promoted more negative and ambiguous emotions than the creative intervention.

Additionally, Table 2 shows the differences between traditional and creative sessions on the psychological variables. Significant differences were found between creative and traditional sessions on the satisfaction of perceived competence (*p*-value = 0.04), social relationships (interactions and cooperative skills) (*p*-value = 0.03) and adherence to this sport (*p*-value = 0.02). All these differences are favorable to the creative session.

Table 3 shows the differences in the type of emotions induced by a traditional and a creative session. Participants showed significantly more humor (a positive emotion) (*p*-value = 0.04) in the creative session than in the traditional session. In addition, the traditional session induced significantly more surprise (an ambiguous session) (*p*-value = 0.01) than the creative session.

### 3.2. Qualitative Results

#### 3.2.1. Participants´ Opinions

After the creative session, participants were asked what they liked about the class more. Positive responses emerged. In this regard, probably the group-based work conducted during the session made one participant express:


*“I like it especially when we practice it. (Referring to phase 3 of the session where they have to practice the choreography with their classmates)”*


When the sports professional asked, why did they like the session? One participant told him that:


*“…because we look like skaters.”*


This opinion could be derived from the fact that they chose the basic elements that they wanted to perform, and they felt like the protagonists. This can be reinforced with the opinion of another participant, who manifested that:


*“…because we have discussed among the classmates, we have invented it ourselves and we have performed it ourselves.”*


However, there are other comments that evidenced the lack of habit when making decisions and planning. In this regard, after the traditional session, one girl affirmed that:


*“I like it (traditional session) better than the other (referring to the creative session), because I find it more difficult when we have to do it ourselves.”*


In contrast, when the sports professional asked what they liked best: to design the choreographies or to do the imposed ones? They responded that:


*“I prefer to design the choreography.”*


#### 3.2.2. Sports Professional’s Opinion

The sports professional stated that the technical level of the basic elements executed by the skaters was similar in both sessions:


*“I have observed a similar technical level of the skaters in both choreographies.”*


However, he ensured that when the skaters designed their own choreographies, they tried to avoid the use of more complex elements.


*“The artistic figure most executed by the skaters is the “dancer” (of less complexity), and they try to avoid jumps and turns (of greater complexity), despite knowing how to execute them.”*


He also recognized that skaters had difficulties when it came to designing choreography themselves, which is reflected in the design of short variations:


*“The choreographies designed by the skaters themselves are usually of short duration and with little variation of elements/sequences.”*


Despite the difficulties caused by the lack of habit, he affirmed that when the skaters performed choreographies designed by him, after some time, the skaters usually asked if they could change activities. On the other hand, when the choreographies are designed by themselves, this does not happen:


*“Sometimes, the skaters themselves ask to change activities when they perform choreographies designed by me because they get tired and bored. This aspect does not occur when they are the ones who design their choreography.”*


## 4. Discussion

Sierra-Díaz, González-Víllora, Pastor-Vicedo and López-Sánchez [19] encouraged further investigations to demonstrate that innovative and creative strategies have positive effects on psychosocial variables and sports adherence in contrast to traditional strategies [19]. Therefore, the present study aimed to investigate the impact of a creative and a traditional session with different psychological parameters (BPNs, MIFA-sports adherence and emotions) on figure roller-skaters. Based on the main results, Hypothesis 1 was partially fulfilled because the creative session obtained significantly higher satisfaction in the needs of perceived competence and social relationships than the traditional intervention. Although, no significant differences were found in the autonomy need. Hypothesis 2 was accepted because the creative session also exhibited significantly higher adherence to roller skating. However, Hypothesis 3 was rejected because there were no significant differences between both sessions in emotions. Although, participants showed significantly more humor (a positive emotion) and less surprise in the creative session than in the traditional session.

Previous studies have shown that active, creative and innovative pedagogical models are the most beneficial for sports practice [18,19,33]. Our results support this hypothesis, since the creative session led to a significant increase in the satisfaction of competence and social relationships in BPNs compared to the traditional session. In the same line, Sierra-Díaz et al. [19] reported that method-based practices are also suitable pedagogical methods for increasing sports competence and self-determined motivation compared to traditional methods. This is a relevant finding, especially in sports with a high level of technical commitment (like skate-rolling), as practitioners perceive themselves as less competent when engaging in skill-based exercises, which are characteristics of traditional methods [19]. Additionally, a previous study indicated that perceived competence is the main need to address the promotion of physical sports activities [34]. This could also explain the results obtained in the present study, where a creative-based session led to significantly higher adherence than a traditional roller skating session.

In addition, our results also showed that a creative session significantly led to higher satisfaction in the social relationships of participants. This finding is relevant since improvements in social relationships could be related to enhanced physical activity and health promotion in children and youth [35]. Furthermore, social media and smartphones have deteriorated socialization and mental health in children and adolescents [36]. In light of the results observed in this exploratory pilot study, creative-based interventions could be used to lessen the negative impacts of social media on the social health of children and adolescents.

Creativity is characterized as a process involving an individual’s emotional aspects [37]. In the present study, participants showed significantly more humor (a positive emotion) and less surprise in the creative session than in the traditional session. Greco and Ison [38] studied the influence of positive emotions on inter-personal cognitive problem-solving skills in 7- and 9-year-old children. Despite not reporting significant differences, these authors stated that the patterns of measures indicated that children who experienced emotions were more prone to present assertive solution alternatives, anticipate positive consequences and make assertive decisions compared to children presenting fewer positive emotions. Moreover, after the traditional session, participants reported significantly higher levels of surprise (an ambiguous emotion). These results might be explained by the type of activities [39] and information [40] provided during the traditional session. In this regard, during the traditional session, participants played a passive role and followed the guidelines (prescriptive/descriptive feedback) of the sports professional. For these reasons, it is necessary to propose a creative pedagogy to improve the creative and psychological development of children and youth (school and out-of-school) and, in turn, to promote sports adherence and health. For this reason, sports professionals and physical education teachers must be trained in the use of active, creative and innovative methodologies.

This pilot study has some limitations that should be acknowledged. First, given the nature of this study, only one session for creativity and traditional interventions has been conducted. Thus, longitudinal or randomized controlled trials with a larger sample size should corroborate our results. Second, in this study, creativity was introduced in roller skating sessions. It would be interesting to study the impact of creativity in other contexts, such as education or other types of sports. Third, the feedback provided during each session was not controlled. Nurjan [41] stated that feedback is a stimulus that elevates creativity, providing challenges in the form of problems and inspiring questions that inspire children’s curiosity. Therefore, future research should also study the feedback provided by sports professionals during the sports process through qualitative analysis.

## 5. Conclusions

A creative session in recreational figure roller-skating might be more appropriate for the training of this sport than a traditional session because it leads to greater satisfaction of the perceived competence, social relationships need and greater adherence to this sport. In addition, participants showed significantly higher levels of humor (a positive emotion) and less surprise in the creative session than in the traditional session. Therefore, sports professionals and physical education teachers should use innovative and creative interventions in which athletes/students explore and experiment to develop their creativity. Future longitudinal studies should corroborate these results after conducting creative-based interventions in sports and educational contexts.

## Figures and Tables

**Table 1 ijerph-19-11407-t001:** Characteristics of creative and traditional intervention.

Session Aspects	Creative Intervention	Traditional Intervention
Choreography type	Choreographies created by the participants themselves	Choreography created by the sports professional
Skaters grouping	Sub-groups according to their interests and friendships	One group only (individual work)
Teaching style(s) [28]	Socializer; cooperative creative; synectics	Traditional: modification of direct control
Role of the sports professional	Guides, encourages and fosters participants’ creativity, without control and discipline	Plans, explains and demonstrates the activity to be performed, exercising moderate control and discipline
Role of the skaters	Active: groups work independently, dictating their own rules and acquire a high cognitive participation grade	Passive: execute the activities according to the guidelines set by the sports professional

**Table 2 ijerph-19-11407-t002:** Differences in psychological variables were analyzed according to the intervention type.

Variable	Dimension	Intervention	*M*	*SD*	*Z*	*p*-Value	*ES*	Comparison
BPNs	Autonomy	Traditional	3.98	1.09	−1.61	0.13	0.46	
		Creative	4.47	0.58				
	Competence	Traditional	3.93	1.07	−2.04	0.04 *	0.59	TI < CI
		Creative	4.55	0.58				
	Relationships	Traditional	4.04	0.90	−2.15	0.03 *	0.62	TI < CI
		Creative	4.52	0.78				
MIFA (sports adherence)		Traditional	4.05	0.92	−2.40	0.02 *	0.69	TI < CI
		Creative	4.57	0.62				
Emotions	Positive	Traditional	6.79	2.56	−1.54	0.13	0.45	
		Creative	8.08	1.74				
	Negative	Traditional	1.24	1.33	−0.62	0.56	0.18	
		Creative	1.04	1.17				
	Ambiguous	Traditional	6.00	1.97	−1.61	0.11	0.46	
		Creative	3.84	1.90				

Note: *M* = mean; *SD* = standard deviation; *ES* = effect size; BPNs = basic psychological needs; MIFA = measure of intentionality to be physically active; TI = traditional intervention; CI = creative intervention; * *p*-value < 0.05.

**Table 3 ijerph-19-11407-t003:** Differences in the type of emotion induced by the traditional and the creative session.

Emotions		Traditional Intervention		Creative Intervention					
		*M*	*SD*	*M*	*SD*	*Z*	*p*-Value	*ES*	Comparison
Positive emotions	Joy	8.70	1.95	9.60	1.26	−1.54	0.14	0.44	
	Humor	4.20	4.59	7.60	3.37	−2.04	0.04 *	0.59	TI < CI
	Love	5.50	4.88	5.30	4.99	−0.273	0.78	0.08	
	Happiness	9.10	1.91	9.60	1.26	−1.07	0.31	0.31	
Negative emotions	Fear	0.60	1.35	0.50	0.97	−0.36	0.74	0.10	
	Anger	0.40	1.26	0.30	0.95	−0.29	0.77	0.08	
	Rejection	1.60	3.20	0.80	2.20	−0.62	0.55	0.18	
	Sadness	0.30	0.67	0.20	0.42	−0.57	0.57	0.17	
	Shame	2.40	3.72	2.60	3.34	−0.44	0.66	0.13	
	Anxiety	2.00	4.22	1.30	2.58	−0.64	0.55	0.18	
Ambiguous emotions	Hope	5.60	4.27	5.00	4.57	−0.538	0.59	0.15	
	Compassion	3.80	4.34	0.70	1.34	−0.81	0.53	0.23	
	Surprise	8.60	1.90	5.70	3.80	−2.55	0.01 *	0.74	TI > CI

Note: *M* = mean; *SD* = standard deviation; *ES* = effect size; TI = traditional intervention; CI = creative intervention; * *p*-value < 0.05.

## Data Availability

Data will be available upon reasonable request from the corresponding author.

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
