# Peer review of "Creativity in Recreational Figure Roller-Skating: A Pilot Study on the Psychological Benefits in School-Age Girls"

_ijerph, 2022, doi:10.3390/ijerph191811407_

Round 1

Reviewer 1 Report

This study involved just one experience of a creative roller skating endeavor and one of the non-creative endeavor, in that order.  The fact that there were differences in the reported experience of the 12 participants is attributed to the one being creative and the other not.  This is quite plausible, but as there was only one experience of each type, this could have been an order effect or could have been due to anything else that happened on one day versus the other.  The researchers should have held more than one day of each type of endeavor, to rule out alternative explanations.  

Given the brief nature of this study, this report is much too long.  There is lots of irrelevant and unnecessary information presented in the introduction and discussion sections. For example, it is not necessary to say what "creativity" is or "emotion" is, and what is said is not helpful.  The English is very poor, to the point of being difficult to understand in places.  There is no discussion of the limitations of this study.

The statement that the inclusion criteria included "to have a contraindication for physical exercise," line 102, is, I assume, a typo.  

In line 130, what is "ira"?

In line 137-138 what does it mean to say that "The order of sessions was randomized"?  It's impossible to randomize order when there was only one session of each type and all participants did each session together.

The graphs here are not necessary or helpful.  A table of means and standard deviations would be much easier to see, and the inferential statistics could be in the same table.

I could imagine this published as a very brief report, making it clear that this is an exploratory study and has the limitation that there was only one experience with each type of roller skating session.

Author Response

Authors´response: Thank you for all your constructive and valuable feedbacks. After incorporating all your suggestions, we truly believe that the quality of the manuscript has been significantly improved.

This study involved just one experience of a creative roller-skating endeavor and one of the non-creative endeavor, in that order.  The fact that there were differences in the reported experience of the 12 participants is attributed to the one being creative and the other not.  This is quite plausible, but as there was only one experience of each type, this could have been an order effect or could have been due to anything else that happened on one day versus the other.  The researchers should have held more than one day of each type of endeavor, to rule out alternative explanations.  

Authors´response: Thank you for your all your constructive and valuable feedback. We have included these aspects as limitations.

Given the brief nature of this study, this report is much too long.  There is lots of irrelevant and unnecessary information presented in the introduction and discussion sections. For example, it is not necessary to say what "creativity" is or "emotion" is, and what is said is not helpful.  The English is very poor, to the point of being difficult to understand in places.  There is no discussion of the limitations of this study.

Authors´response: Thank you for your valuable feedback. We have modified both the introduction and the discussion to be more direct and clearer. Also, the English has been reviewed and rewritten when necessary. A limitation section has been included in the discussion.

The statement that the inclusion criteria included "to have a contraindication for physical exercise," line 102, is, I assume, a typo.  

Authors´response: Thank you for pointing us this typo. It has been removed.

In line 130, what is "ira"?

Authors´response: Thank you again for pointing us this typo. It has been replaced by anger.

In line 137-138 what does it mean to say that "The order of sessions was randomized"?  It's impossible to randomize order when there was only one session of each type and all participants did each session together.

Authors´response: We see your point. Thus, we have rewritten the sentence to improve clarity. We wanted to say that the decision of whether to do the creative or traditional session first was made by lottery.

The graphs here are not necessary or helpful.  A table of means and standard deviations would be much easier to see, and the inferential statistics could be in the same table.

Authors' Response: Thank you for your recommendation. We have replaced Figure 1 and Figure 2 with two tables (Table 1 and Table 2).

I could imagine this published as a very brief report, making it clear that this is an exploratory study and has the limitation that there was only one experience with each type of roller-skating session.

Authors´response: Thank you for your suggestion. We totally agree with your opinion, thus, we have included in the title that it is an exploratory study as well as included limitations in the discussion.

Reviewer 2 Report

This paper presents and inspiring and creative project, involving 12 young female roller skaters, comparing two sessions at the end of a season. In the first session, the skaters created their own choreography in smaller groups, whereas in the second they were taught a choreography, without being involved in its creation. 

The authors conclude that "A creative session in recreational figure roller-skating is more appropriate for the training of this sport than a traditional session because it leads to greater satisfaction of the perceived competence, social relationships need, and greater adherence to this sport."

However, I am not convinced that their limited study is sufficient to support this conclusion. For a start, changes are known to be motivating in themselves, and very little information is given about their normal training sessions. I think that this creative project deserves a presentation that better reflect the qualitative aspects. A lot of the text is spent on general introductions and discussions, with only brief presentations of the results.

For a start, the statistics itself could be presented in a much more informative way than in Figures 1 and 2, by displaying 12 individual responses rather than averages and (presumably) standards deviations. The authors may also comment about the bars reaching above 5, which was presumably the maximum value.

I was unable to find how long each session was, and if different teachers carried out different lessons. 

I would have loved to see some comments from the skaters complementing the statistics. I also miss any discussion about the level of technique displayed in the two sessions. Including descriptions of the choreographies created by the particpants would also make the paper more interesting.

There are some strange sentences, e.g.

line 98 "11 months, from October to May 2022" : Surely, this cannot be more than 8 months

line 102-103 (2) to answer correctly to the questionnaires; (3) to have a contraindication for physical exercise.: Both these points need elaborating. I do not see the contraindications discussed further in the paper.

137-140: I do not understand if the order was randomized or if the first session was always the creative one.

164 (and several other places) I think that "female skaters" should be replaced by "participants" (unless the sentences refer to female skaters in general)

292-299 The authors need to reread and revise the concluding section. The combination "more positive ... no significant differences ... more negative" seems to contradict itself.

I thus find that this manuscript needs a major revision before publication and encourage the authors to share more of the qualitative data that they must have available. 

Author Response

Authors´response: Thank you for all your constructive and valuable feedbacks. After incorporating all your suggestions, we truly believe that the quality of the manuscript has been significantly improved.

This paper presents and inspiring and creative project, involving 12 young female roller skaters, comparing two sessions at the end of a season. In the first session, the skaters created their own choreography in smaller groups, whereas in the second they were taught a choreography, without being involved in its creation. 

The authors conclude that "A creative session in recreational figure roller-skating is more appropriate for the training of this sport than a traditional session because it leads to greater satisfaction of the perceived competence, social relationships need, and greater adherence to this sport."

However, I am not convinced that their limited study is sufficient to support this conclusion. For a start, changes are known to be motivating in themselves, and very little information is given about their normal training sessions. I think that this creative project deserves a presentation that better reflect the qualitative aspects. A lot of the text is spent on general introductions and discussions, with only brief presentations of the results.

Authors´response: Thank you for your constructive and valuable feedback. We totally agree that conclusion should be improved due to the study design employed in this article. Thus, we have moderated the conclusion as well as rewritten some part of it in order to improve clarity.

Moreover, introduction and discussion have been modified in order to be more clear, precise and direct.

For a start, the statistics itself could be presented in a much more informative way than in Figures 1 and 2, by displaying 12 individual responses rather than averages and (presumably) standards deviations. The authors may also comment about the bars reaching above 5, which was presumably the maximum value.

Authors' Response: Thank you for your recommendation. We have replaced Figure 1 and Figure 2 with two tables (Table 1 and Table 2). We truly believe that now the results are much more clearer than before.

I was unable to find how long each session was, and if different teachers carried out different lessons. 

Authors´response: Thank you for pointing us this typo. Both sessions lasted one hour. In addition, the two sessions were conducted by the same teacher. All these data have been added in the manuscript.

I would have loved to see some comments from the skaters complementing the statistics.I also miss any discussion about the level of technique displayed in the two sessions. Including descriptions of the choreographies created by the participants would also make the paper more interesting.

Authors´response: Thank you for this interesting suggestion. We have included a small qualitative result section in the manuscript. Initially, we focused on a quantitative approach. Nevertheless, you were totally right and these results could complement the quantitative results.

There are some strange sentences, e.g.

line 98 "11 months, from October to May 2022": Surely, this cannot be more than 8 months

Authors´response: Corrected.

line 102-103 (2) to answer correctly to the questionnaires; (3) to have a contraindication for physical exercise.: Both these points need elaborating. I do not see the contraindications discussed further in the paper.

Authors´response: Thank you for your suggestion. We have modified the inclusion criteria in order to improve clarity. Regarding the third inclusion criteria, following the suggestion of another reviewer has been removed.

137-140: I do not understand if the order was randomized or if the first session was always the creative one.

Authors´response: We see your point. Thus, we have rewritten the sentence to improve clarity. We wanted to say that the decision of whether to do the creative or traditional session first was made by lottery.

164 (and several other places) I think that "female skaters" should be replaced by "participants" (unless the sentences refer to female skaters in general)

Authors´response: Thank you for your suggestion. It has been replaced were appropriate.

292-299 The authors need to reread and revise the concluding section. The combination "more positive ... no significant differences ... more negative" seems to contradict itself.

Authors´response: Thank you for your suggestion. It has been rewritten.

I thus find that this manuscript needs a major revision before publication and encourage the authors to share more of the qualitative data that they must have available. 

Authors´ response: Thank you for your feedback. 

Round 2

Reviewer 1 Report

This draft is considerably improved over the previous one.  I like the addition of quotes illustrating the participants' and instructor's views of the two types of sessions. The tables are a great improvement over the graphs.  I still think this could be written in a more concise way, as a considerably shorter article, but that is an issue for the editor to decide.  The abstract is, in my opinion too long; most journals would not allow such a long abstract even for a major article, let alone for a  pilot study such as this.  The English is improved, but still needs the work of a good editor.  By the way, "Europa" (line 57, p 2) is one of the moons of Jupiter.  I don't think that's what you meant.  Also, in the title, calling it both a pilot study and exploratory study is redundant.  Choose one or the other.

Author Response

We would like to express our gratitude to reviewer 1 for the time in reviewing our manuscript.

Reviewer’ note: This draft is considerably improved over the previous one.  I like the addition of quotes illustrating the participants' and instructor's views of the two types of sessions. The tables are a great improvement over the graphs.  I still think this could be written in a more concise way, as a considerably shorter article, but that is an issue for the editor to decide.  The abstract is, in my opinion too long; most journals would not allow such a long abstract even for a major article, let alone for a pilot study such as this.  The English is improved, but still needs the work of a good editor.  By the way, "Europa" (line 57, p 2) is one of the moons of Jupiter.  I don't think that's what you meant.  Also, in the title, calling it both a pilot study and exploratory study is redundant.  Choose one or the other.

Authors’ response: Authors ‘response: Thank you for your positive and constructive feedback. Your comments have helped improve the quality of the manuscript.  You were totally right that the abstract was excessively extensive. In fact, it has more than the 200 words allowed by the journal. Furthermore, the title has been changed and these typos (as well as other detected) corrected.

Reviewer 2 Report

I thank the authors for the clarifications and additions, and think that the paper may now be published

Author Response

We would like to express our gratitude to reviewer 2 for the time in reviewing our manuscript.

Reviewer’ note: I thank the authors for the clarifications and additions, and think that the paper may now be published

Authors’ response: Authors ‘response: Thank you for your positive and constructive feedback. Your comments have helped improve the quality of the manuscript.